# Significant Interplay Between Lipids, Cytokines, Chemokines, Growth Factors, and Blood Cells in an Outpatient Cohort

**DOI:** 10.3390/ijms26167746

**Published:** 2025-08-11

**Authors:** Mats B. Eriksson, Lars B. Eriksson, Anders O. Larsson

**Affiliations:** 1Department of Surgical Sciences, Uppsala University, Uppsala University Hospital, SE-751 85 Uppsala, Sweden; lars.b.eriksson@regiondalarna.se; 2NOVA Medical School, New University of Lisbon, 1099-085 Lisbon, Portugal; 3Department of Medical Sciences, Uppsala University, Uppsala University Hospital, SE-751 85 Uppsala, Sweden; anders.larsson@akademiska.se

**Keywords:** biomarker, blood cells, cardiovascular, cytokine, inflammation, lipid

## Abstract

Cardiovascular disease (CVD) remains the leading global cause of morbidity and mortality, largely driven by atherosclerosis, a chronic inflammatory process involving lipids and immune cells. Although traditional lipid biomarkers such as low-density lipoprotein (LDL) and high-density lipoprotein (HDL) are well-established in CVD risk stratification, the interplay between cytokines, chemokines, growth factors (CCGFs), lipid metabolism, and hematological parameters in non-cardiac populations remains underexplored. We investigated associations between plasma cytokines and lipid-related biomarkers and their relationships with circulating blood cell counts in a cohort of 164 essentially healthy adults aged 18–44 years. CCGF profiling was performed using a proximity extension assay (PEA), and statistical correlations were adjusted for multiple testing using false discovery rate (FDR) correction. The CCGFs that were associated with HDL and apolipoprotein A1 all displayed negative associations. Several pro-inflammatory cytokines, including CCL3, IL-6, and TNFSF10, showed strong positive associations with triglycerides, remnants, non-HDL, and body mass index (BMI). Furthermore, triglycerides and remnants were consistently correlated with elevated leukocyte, neutrophil, and platelet counts. HGF and FGF-21, mainly considered as anti-inflammatory, were positively associated with BMI and negatively associated with HDL, which is compliant with a multitude of actions, depending on the local milieu and the cellular interplay. Our results support the existence of a complex immunometabolic network involving lipids, CCGFs, and blood cells, even in non-diseased individuals. The observed patterns underscore the importance of understanding the intricate cytokine–lipid–cell interactions that may occur in early pathophysiological processes and highlight their potential utility in refining cardiovascular risk assessment beyond traditional lipid metrics.

## 1. Introduction

Cardiovascular disease (CVD), a leading cause of morbidity and mortality, affects more than 500 million people globally, and its prevalence seems to be increasing. Atherosclerotic diseases are the main mediators of CVD. Approximately 50% of CVD deaths are attributed to ischemic heart disease and an additional 25% are caused by ischemic stroke. CVD not only is associated with increased morbidity and mortality, but also confers a substantial economic burden to the health-care system [1,2,3,4,5,6,7].

Atherosclerosis is an inflammatory disease of the arterial intima, where the balance between pro-inflammatory and anti-inflammatory mechanisms is crucial for the clinical outcome. Intimal infiltration and modification of plasma lipoproteins, particularly LDL, and their uptake by macrophages are key events in the development of atherosclerosis. Arterial LDL accumulation is an important step in atherosclerotic plaque formation. Low-density lipoprotein (LDL) concentrations in the arterial intima far exceed concentrations in other connective tissues. Genetic variations and host immune–inflammatory responses can modulate the pro-atherogenic effect of elevated LDL-cholesterol. High-density lipoprotein (HDL) particles have cardiovascular-protective effects, primarily attributed to their ability to transport cholesterol to the liver, where it can be excreted or converted into bile acids. HDL particles also have antioxidant and anti-inflammatory roles. There is a causal relationship between triglycerides, triglyceride-rich lipoproteins, and their remnants in atherosclerotic disease, especially in patients with obesity, metabolic syndrome, diabetes, and chronic kidney disease [8,9,10,11].

Pro-inflammatory cytokines are linked to several types of CVDs. The most important cytokines in this respect include interleukin-6 (IL-6), tumor necrosis factor (TNF) alpha, and the interleukin-1 (IL-1) family. Inflammation is involved in numerous pathophysiological processes such as oxidative stress and calcium-related signaling events that may facilitate leukocyte–endothelial cell interactions, indicating the dynamic nature of pro-inflammatory cytokines in several CVDs. As part of the inflammation-induced endothelial dysfunction, there is increased permeability to lipoproteins, leading to deposition in the subendothelial space, leukocyte migration, and platelet activation. Once inside the arterial wall, LDL-cholesterol undergoes oxidation, while triglyceride-rich lipoproteins and remnant lipoproteins exert pro-inflammatory effects. Furthermore, psychological stress may increase the risk of cardiovascular disease, mediated by the release of inflammatory cytokines. Furthermore, inflammation is a link between aging and cardiovascular disease, as aging presents with systemic low-grade chronic inflammation and elevated concentrations of mediators such as IL-6, TNFα, and C-reactive protein (CRP) [12,13,14].

High levels of erythrocytes are thrombogenic, which may not only be due to impaired blood flow, since erythrocytes may also up-regulate IL-8 mRNA even if IL-6 and VEGF mRNA expression appears down-regulated. IL-8 plays a crucial role in neutrophil recruitment, which may induce potent cytotoxic effects through neutrophil extracellular trap (NET) formation and the release of proteolytic enzymes. Erythrocytes from healthy individuals regulate immune cell activity and bind more than 50 cytokines, hereby playing an active role in cytokine signaling and regulation. Neutrophil cells have important functions, including not only cardiovascular inflammation and repair, but also atherogenesis, plaque destabilization, and plaque erosion. Activated platelets are highly involved in inflammatory processes, where they express a plethora of pro- and anti-inflammatory molecules that attract circulating leukocytes. Furthermore, platelets can directly influence adaptive immune responses. Significant associations between cytokines in saliva and peripheral blood cells were recently published by our group [15,16,17,18,19,20,21,22].

The aims of the study were to (1) investigate associations between cytokines, chemokines, and growth factors (CCGFs) and selected biomarkers of cardiovascular disorders in a non-cardiac cohort; and (2) explore potential relationships between these biomarkers and circulating blood cells in the same cohort.

## 2. Results

### 2.1. Patient Characteristics

The cohort consisted of 164 individuals (53 males). The mean age was 29 years and the range was 18–44 years.

Prevalences of the CCGFs analyzed in our cohort are displayed in Table 1.

### 2.2. Cytokine Values Below the Assay’s Standard Curve Limits

Table 2 displays the number of results below the lowest standard point for each of the Olink markers. There were no values exceeding the assay’s highest standard points.

### 2.3. Biomarkers vs. Cytokines

All significant associations, after adjustment for multiplicity, between CCGFs and other biomarkers are shown in Appendix A.

HDL exhibited 12 significant associations out of the quantified CCGFs (Figure 1), all of them negative. CCL3 was the one that was most negatively associated with HDL, whereas TNFRSF9 was the cytokine that exhibited the least expressed negative association with HDL.

Apolipoprotein A1 was negatively associated with nine cytokines (Figure 2). TNFSF10 was the cytokine that was the least negatively associated, whereas PLAU was the most negatively associated one.

Creatinine was positively associated with 11 CCGFs (Figure 3), and eGFR_creatinine_ (Figure 4) was associated with 3 CCGFs. FGF-19 was the second-most associated with creatinine, but was negatively associated with eGFR_creatinine_.

Albumin was associated with nine CCGFs (Figure 5). Five of these associations were positive.

Apolipoprotein B was positively associated with seven CCGFs (Figure 6), where CCL3 showed the strongest association and TNFSF10 the weakest one.

Triglycerides were associated with eight of the analyzed CCGFs. The strongest associations between triglycerides and CCGFs were noted for FGF-21, followed by CCL3 (Figure 7).

Remnants (=Non-HDL–LDL) exhibited several similarities with and were almost identical to the triglycerides, except for the presence of CDCP1 in remnants (Figure 8).

Non-HDL was equally and strongly associated with both CCL3 and CDCP1, followed by TNFSF10 (Figure 9).

LDL was merely associated with both TNFSF10 and CDCP1.

Total cholesterol was not associated with any of the assessed CCGFs.

Age exhibited eight associations. The only CCGF that was positively associated with age was Flt3L, whereas the most negative association with age was noted for IL-18R1 (Figure 10).

Both weight and body mass index displayed complex association patterns, comprising sixteen and fourteen associations, respectively. The most expressed associations for both of them were with IL6 (Figure 11 and Figure 12, respectively).

Gender was associated with fifteen CCGFs (Figure 13), with multiple inter-cytokine interactions. All associations were negative, and TNFSF10 was the cytokine that exhibited the strongest negative association with gender.

### 2.4. Biomarkers vs. Hematology

All significant associations, after adjustment for multiplicity, between biomarkers and hematological data are displayed in Appendix A.

Both remnants and triglycerides were strongly associated with the erythrocyte count, leukocytes, platelets, hemoglobin, erythrocyte volume fraction, neutrophils, and mean corpuscular volume (negative associations). Except for the fact that weight was associated with mean corpuscular hemoglobin concentration (MCHC), there were definite similarities between this biomarker and both remnants and triglycerides. BMI was associated with leukocytes, neutrophils, platelets, and the erythrocyte count. Total cholesterol was associated with erythrocyte count, platelets, and mean corpuscular hemoglobin (MCH), whereas total cholesterol was not associated with any of the cytokines that were quantified.

eGFR_creatinine_ was not associated with any of the hematological data.

### 2.5. CCGF Correlations

Correlations between the proteins in plasma are shown in three correlation matrices, displaying Spearman rank values for each association in the entire cohort, among men, and among women, respectively (Appendix A). Values in red denote *p* < 0.05.

## 3. Discussion

Although cytokines can exert diverse effects depending on the biological context, most exhibit predominantly pro- or anti-inflammatory functions. From an evolutionary perspective, it would have been disadvantageous to develop signaling molecules that are not, under specific conditions, beneficial to the host. Nevertheless, inflammatory responses are often Janus-faced, reflecting both protective and potentially harmful roles, and their specific regulatory features may have shifted over time as new species have emerged.

Independent of their predominant pro- or anti-inflammatory functions, all cytokines assessed in this study were negatively correlated with apolipoprotein A and HDL levels. From a quantitative aspect, TNFSF10 appears to be the most important cytokine in our study. This pro-inflammatory cytokine has been linked to improved survival in cancers with high tumor-associated macrophage content, which may reflect its capacity to induce cell death or alternatively to activate survival-promoting pathways depending on the tumor context [23]. Furthermore, TNFSF10 was the cytokine that had the strongest negative association with apolipoprotein A1 and exhibited the most positive association with LDL.

CCL3, the second-most frequently found cytokine in our cohort, is an inflammatory cytokine, secreted by monocytes and macrophages, having a fundamental role when tumor-associated macrophages have an impact on tumor development [24]. CCL3 was negatively associated with apolipoprotein A1 and HDL, but positively associated with apolipoprotein B, BMI, non-HDL, remnants, triglycerides, and weight.

TNFSF11 is an inflammatory cytokine that correlates with age-related macular degeneration [25]. TNFSF11 was significantly negatively associated with apolipoprotein A1, gender, and HDL. Positive associations were noted between TNFSF11 and apolipoprotein B, BMI, non-HDL, and weight.

IL6, conventionally seen as a pro-inflammatory cytokine, was positively associated with risk factors for cardiovascular disease, e.g., BMI, eGFR_creatinine_, remnants, triglycerides, and weight, but not with increasing age, which in a previous study has been associated with elevated levels of IL6 [26]. IL6 was negatively associated with HDL, indicating an immunometabolic interplay that affects the risk of cardiovascular events [27,28]. The inflammatory member of the tumor necrosis factor ligand superfamily, TNFRSF9 [29], was negatively associated with age, apolipoprotein A1, gender, and HDL, but positively associated with creatinine, which overall may suggest a negative impact of this cytokine.

The inflammatory cytokine Flt3L was associated with age, a finding in agreement with a previous study in 94 strictly healthy volunteers, aged 18–80 years old [30].

The two most frequent cytokines, which mainly have anti-inflammatory properties, noted in our cohort were HGF [31] and FGF-21 [32], respectively. HGF was positively associated with BMI, non-HDL, and weight, whereas HGF was negatively associated with gender, and, somewhat surprisingly, also negatively associated with HDL. FGF-21 exhibited a similar pattern, were BMI, remnants, triglycerides, and weight were positively associated, while gender and HDL, once again, were negatively associated.

IL-10RB [33], FGF-19 [34], and COL18A1 (VEGFA) [35], having at least partly anti-inflammatory properties, exhibited three significant associations each. Both IL-10RB and COL18A1 were positively associated with remnants as well as triglycerides. IL-17C exhibited the strongest Spearman correlation against creatinine in our cohort. This cytokine is known to have a pathogenic role in renal damage, and IL-17C neutralization protects the kidney against both acute and chronic injury [36,37]. Thus, it is remarkable that even in an essentially healthy cohort, significant associations between potentially nephrotoxic cytokines and creatinine were observed. FGF-19 showed the second-strongest positive association with creatinine, while also being negatively associated with eGFR_creatinine_. In this context, it is noteworthy that high levels of FGF19 were found in non-diabetic patients with chronic kidney disease [38]. A bidirectional relationship exists between the heart and the kidneys, whereby dysfunction in one organ system can lead to dysfunction in the other [39]. Cytokines play a crucial role in the development of this cardiorenal syndrome [39].

The most striking finding when biomarkers were associated with hematological data was that WBC (especially neutrophil cells) and TPK were frequently related to BMI, remnants, triglycerides, and weight. This is not surprising, since WBC, neutrophils, and platelets are all related to triglycerides [40,41]. Leukocytes, in particular neutrophils, and platelets secrete inflammatory cytokines, which are related to chronic low-grade inflammation of adipose tissue [42]. This is a complex, multifaceted process, where adipocytes secrete inflammatory adipokines, cytokines, and chemokines. Leukocytosis and increased platelet counts are driven by this inflammatory state [42,43,44]. In this context it is noteworthy that statins significantly reduce plasma levels of CRP, as well as several pro-inflammatory cytokines, thereby exerting beneficial effects by lowering the levels of these inflammatory markers [45], a finding in alignment with our results on the associations between CCGFs and blood lipids.

Leukocytes and especially neutrophil cells have a distinct role in this context, as increased numbers of white blood cells and neutrophils will also increase the formation of neutrophil extracellular traps (NETs). NETs are web-like structures composed of DNA, histones, and granule proteins that are released by activated neutrophils. This process, known as NETosis, is a distinct form of cell activation that allows neutrophils to trap and kill pathogens extracellularly. While NETs serve a protective role in host defense, dysregulated or excessive NET formation has been implicated in the pathogenesis of various diseases, particularly in the cardiovascular and renal systems [46,47].

NETs contribute to several aspects of cardiovascular pathology. NETs promote plaque formation and destabilization in atherosclerosis by activating macrophages and endothelial cells. They serve as a scaffold for platelet adhesion and coagulation factor activation, thus contributing to thrombus formation.

NET components such as histones and myeloperoxidase can damage cardiomyocytes and propagate inflammation post-infarction. In recent years, growing evidence has highlighted the strong link between coagulation and inflammation, leading to the emergence of the concept of immunothrombosis. NETs are found within venous thrombi and facilitate fibrin deposition and coagulation, linking inflammation and thrombosis (termed immunothrombosis).

NLRP3 is a key component of the innate immune system and forms part of the inflammasome that has been implicated in chronic low-grade inflammation associated with metabolic syndrome, including obesity, insulin resistance, and type 2 diabetes [48]. NLRP3 elicits maturation of the cytokines IL-1β and IL-18 [49]. We noted that IL18R1, a receptor for IL-18, is negatively associated with age and gender, but positively associated with Apo B and weight.

This process involves coordinated interactions between leukocytes, platelets, and coagulation factors. Neutrophil extracellular traps (NETs) play a pivotal role by offering a scaffold that promotes platelet activation, thrombin generation, and fibrin deposition, thereby contributing to clot formation. NET formation has been shown to induce inflammation and cardiac injury [46]. NET formation has also been shown to cause kidney injury [47]. In ischemia–reperfusion injury or sepsis, excessive NET formation promotes microvascular thrombosis, endothelial damage, and inflammation, all contributing to renal dysfunction [47]. During NET formation the nuclear and granule membranes break down, allowing the release of proteins stored within the neutrophil [50]. Given their roles in defense and disease, NETs have become targets for novel therapeutic approaches, including anti-inflammatory agents that reduce NETosis.

Furthermore, lipids have signaling roles in platelets and regulate how lipids generated by platelets influence other cells [51,52]. Lipids also play an important role in cell fate decisions during hematopoiesis [52], which may be in alignment with our finding that platelets were also associated with apolipoprotein B and total cholesterol, respectively. We have recently shown that cytokines, in both peripheral blood [21] and human saliva [22], are associated with blood cell counts. These associations between circulating cytokines and peripheral blood cell counts are in agreement with the present paradigm on the complex interrelations between lipids, cytokines, and peripheral blood cells. For example, we noted strong associations between platelet count and IL-6 [21], which, together with our present study, strengthens the postulate that platelets play a crucial role in the inflammatory process and that inter-cytokine interactions across different cytokine families reflect a part of the large amount of crosstalk that is a part of immunologic homeostasis [53].

### Limitations and Strengths

This study has several limitations. It is a single-site study from a region where the majority of the patients were Caucasians. There is a distinct female preponderance among the subjects. Also, the cohort is fairly uniform in age and was clinically evaluated to be essentially healthy, without any acute or systemic severe illness.

The STRING images visualize predicted protein–protein interactions within an integrated network, thus facilitating biological interpretation and hypothesis generation, which might be of translational relevance in future studies.

## 4. Materials and Methods

### 4.1. Population

Patients referred to the Department of Oral and Maxillofacial Surgery at the Falu County Hospital, Sweden, were recruited by LBE and offered to participate in this study, which focused on men and women from 18 to 44 years old, with a bodyweight of 50 to 120kg. Healthy patients, or those with well-compensated systemic disease, were accepted for screening. Inclusion and exclusion criteria and the surgical procedure have previously been described in detail [54,55]. After successful screening, those who had signed a written informed consent form were included.

The cohort is briefly described in Table 3, in compliance with GDPR (EU 2016/679).

### 4.2. Sampling Procedures

Blood samples were collected in vacutainer tubes while the patient was in the supine position. All blood samples were obtained prior to surgery. Complete blood counts were analyzed in EDTA–blood at the clinical laboratory at Falun Hospital. Conventional biochemical analyses, including estimated glomerular filtration rate (eGFR_creatinine_) [56], were performed. Plasma cytokine samples were frozen and stored at −80 °C until analysis.

### 4.3. Ethics

This study was conducted in accordance with the principles of the Helsinki Declaration [57]. Blood sampling was the only difference from clinical routine treatment, which implied a further invasive step. The study was approved by the Swedish Ethical Review Authority (Dnr 2015/378) on 2 December 2015 and registered in the European Union Regulating Authorities Clinical Trials Database (EudraCT) under number 2014-004235-39 on 29 September 2014. Furthermore, the trial was listed on ClinicalTrials.gov with the ID NCT04459377 on 8 July 2020.

### 4.4. Proximity Extension Assay

The proximity extension assay (PEA) was performed using the Proseek Multiplex Inflammation kit [Olink Bioscience, Uppsala, Sweden; (v3024)] [58,59,60]. The PEA seems to reliably reflect protein plasma levels, as compared to conventional assays [61]. In brief, 1 µL of plasma was combined with 3 µL of incubation mix, containing two probes (antibodies conjugated with unique DNA oligonucleotides), and incubated at 8 °C overnight.

Following incubation, 96 µL of extension mix, which included the PEA enzyme and PCR reagents, was added. The samples were then incubated at room temperature for 5 min before undergoing 17 cycles of DNA amplification in a thermal cycler.

A 96.96 Dynamic Array IFC (Fluidigm, South San Francisco, CA, USA) was prepared and primed according to the manufacturer’s guidelines. In a separate plate, 2.8 µL of the sample mixture was combined with 7.2 µL of detection mix, and 5 µL of this solution was loaded into the right side of the primed 96.96 Dynamic Array IFC. Unique primer pairs for each cytokine were loaded into the left side of the array. The protein expression analysis was then performed using the Fluidigm Biomark reader, following the Proseek protocol. The Proseek kit quantified 92 proteins, which are listed in Appendix A along with their full names, UniProtIDs, and corresponding encoding genes.

### 4.5. STRING Images

Cytokine, chemokine, and growth factor interactions and concentration patterns are visualized using images generated from the STRING database [62]. The edge weights are calculated using the STRING database based on the associations found in our study. Protein names are displayed to create interaction networks, and edge thicknesses are indicators of confidence, indicating how likely STRING judges an interaction to be true given the available evidence. The images were exported in high-resolution format for inclusion in figures. This approach enabled a simultaneous representation of both quantitative concentration data and qualitative information about potential interactions.

### 4.6. Statistical Analysis

Coefficients of variation were analyzed using Spearman rank correlations in Statistica (StatSoft, v14; Tulsa, OK, USA). Absolute cytokine values below the lowest standard point were included in the statistical analysis. There were no values exceeding the assay’s standard curve limits when set to the highest or lowest standard value, respectively. To account for the increased risk of false positives due to multiple comparisons, *p*-values were adjusted using the false discovery rate (FDR) approach [63]. Adjusted *p*-values below 0.10, corresponding to an expected FDR of ≤10%, were considered statistically significant.

## 5. Conclusions

This study demonstrates significant associations between circulating cytokines, lipid-related biomarkers, and peripheral blood cell parameters in a healthy, young adult population. Most cytokines were inversely correlated with HDL and apolipoprotein A1, while several positive associations were observed with triglycerides, non-HDL, BMI, and blood cells secreting inflammatory cytokines. These findings underscore the complex interplay between lipids, cytokine signaling, and hematological components, even in a non-cardiac cohort. STRING images provide a visual summary of predicted protein–protein interactions by integrating experimental data, known pathways, and computational predictions. By illustrating interaction networks, they facilitate understanding of complex molecular relationships that extend beyond isolated analyte changes.

Our results may suggest that pre-existing continuous crosstalk between the lipid metabolism and immune inflammatory pathways may lead to subsequent immunometabolic dysregulation, with potential implications for cardiovascular risk assessment.

## Figures and Tables

**Figure 1 ijms-26-07746-f001:**
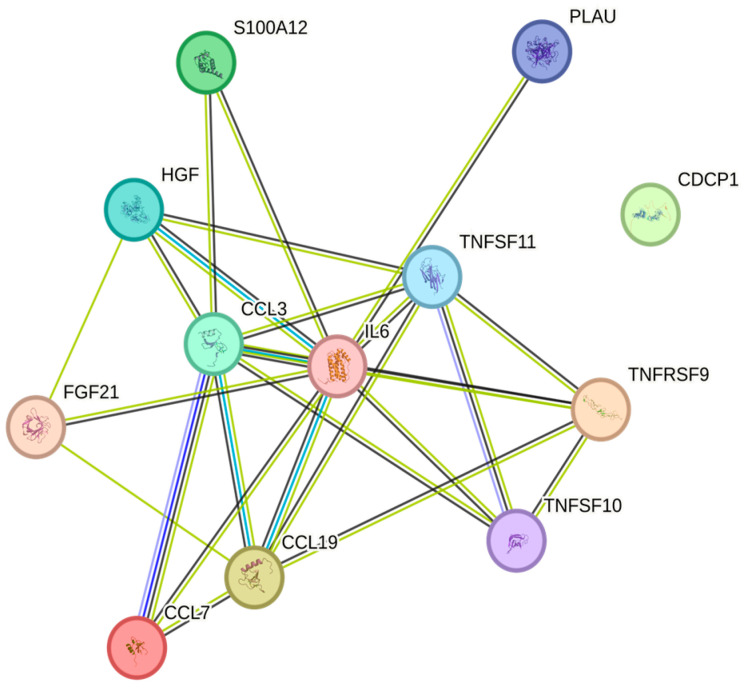
This figure represents a network of interactions among cytokines, demonstrating significant associations between circulating cytokine levels and HDL. The nodes (CCGFs) are connected by edges of varying thickness, indicating confidence of interactions.

**Figure 2 ijms-26-07746-f002:**
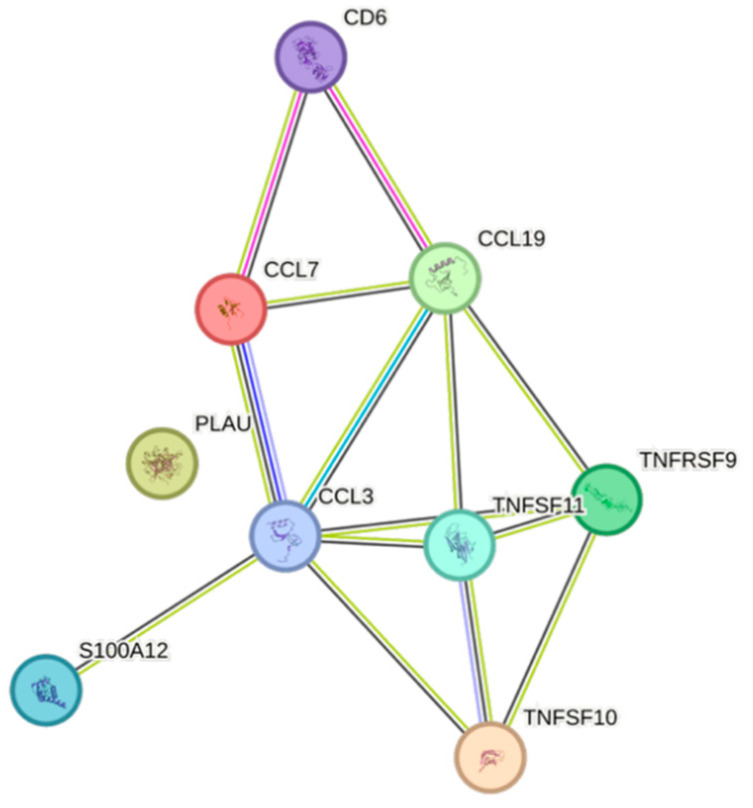
This figure represents a network of interactions among CCGFs, demonstrating significant associations between circulating CCGF levels and apolipoprotein A1. The nodes (CCGFs) are connected by edges of varying thickness, indicating confidence of interactions.

**Figure 3 ijms-26-07746-f003:**
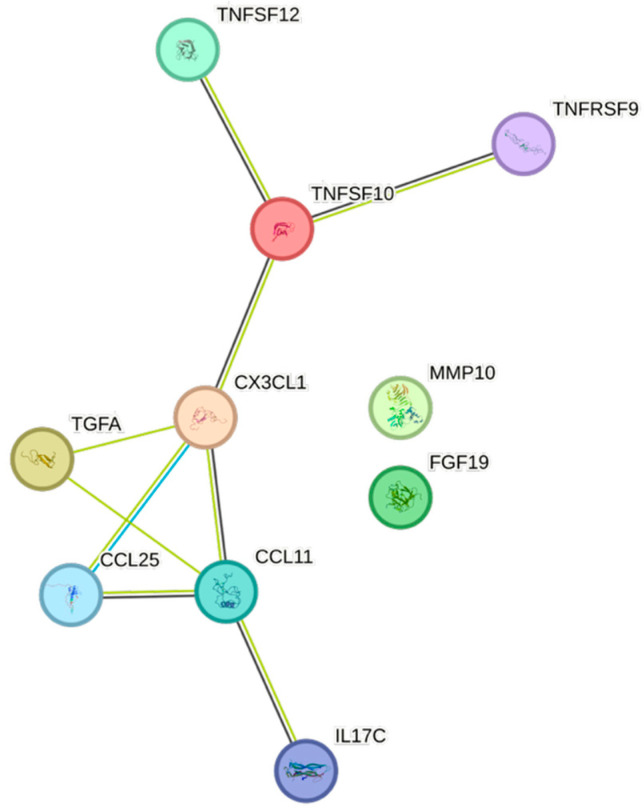
This figure represents a network of interactions among CCGFs, demonstrating significant associations between circulating CCGF levels and creatinine. The nodes (CCGFs) are connected by edges of varying thickness, indicating confidence of interactions.

**Figure 4 ijms-26-07746-f004:**
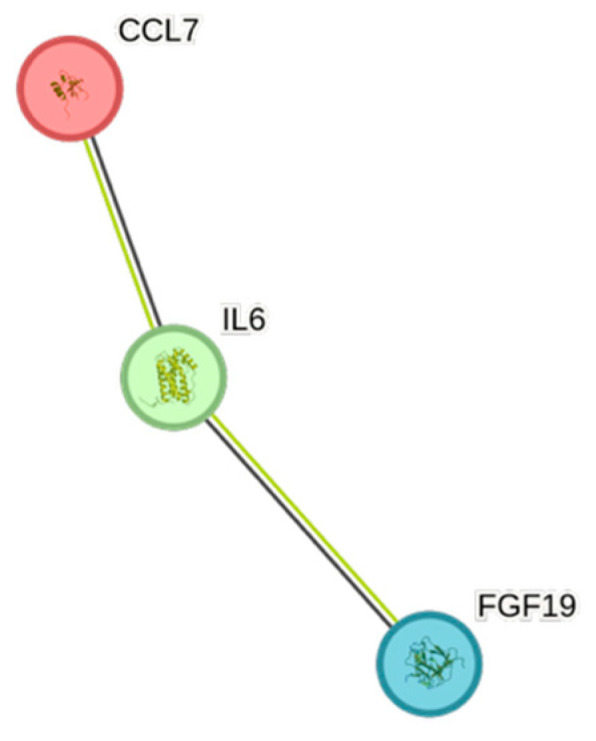
This figure represents a network of interactions among CCGFs, demonstrating significant associations between circulating cytokine levels and eGFR_creatinine_. The nodes (CCGFs) are connected by edges of varying thickness, indicating confidence of interactions.

**Figure 5 ijms-26-07746-f005:**
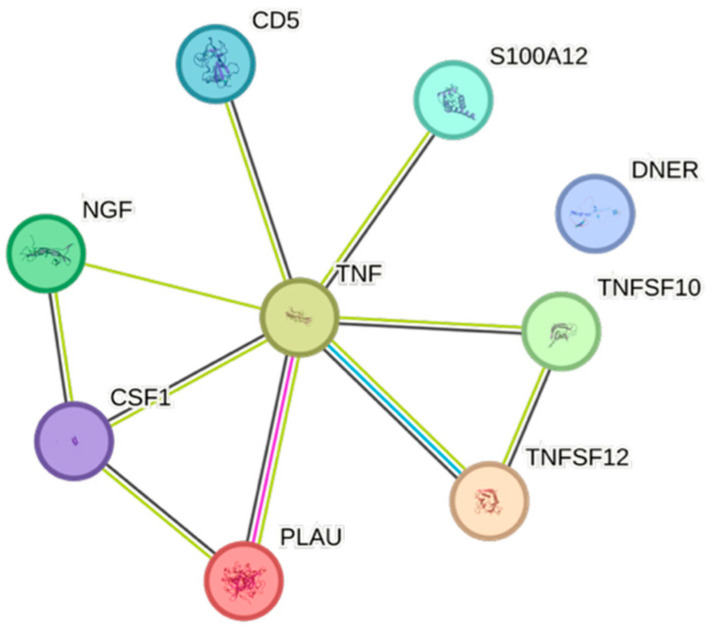
This figure represents a network of interactions among CCGFs, demonstrating significant associations between circulating cytokine levels and albumin. The nodes (CCGFs) are connected by edges of varying thickness, indicating confidence of interactions.

**Figure 6 ijms-26-07746-f006:**
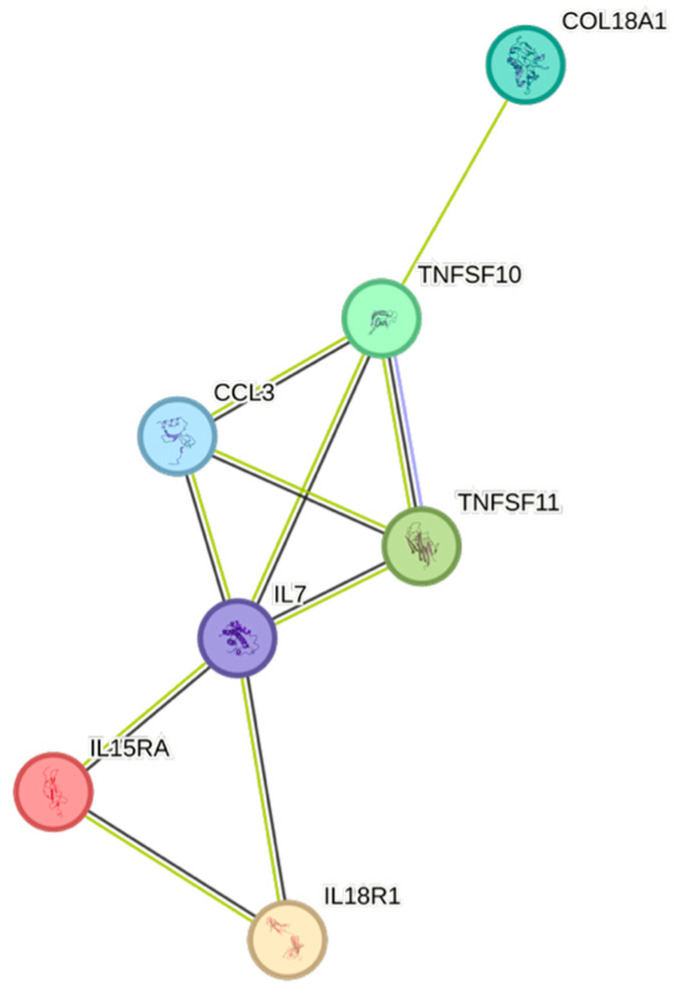
This figure represents a network of interactions among CCGFs, demonstrating significant associations between circulating cytokine levels and apolipoprotein B. The nodes (CCGFs) are connected by edges of varying thickness, indicating confidence of interactions.

**Figure 7 ijms-26-07746-f007:**
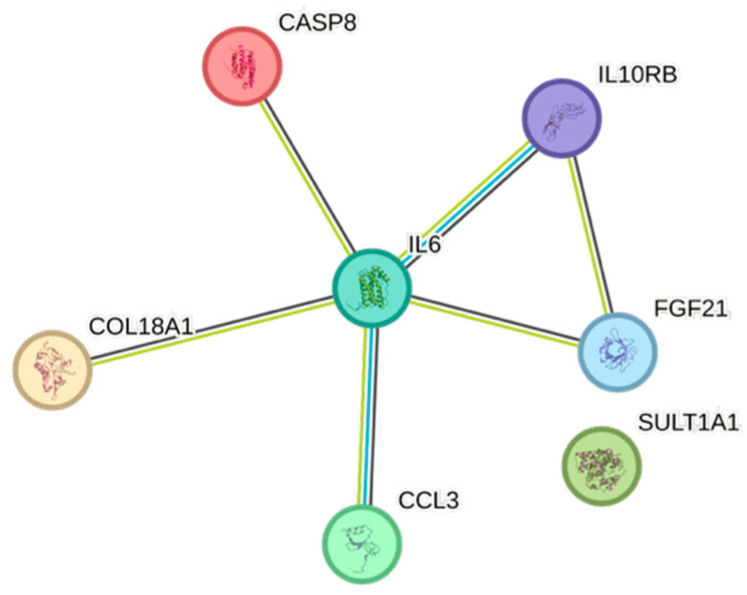
This figure represents a network of interactions among CCGFs, demonstrating significant associations between circulating CCGF levels and non-HDL. The nodes (CCGFs) are connected by edges of varying thickness, indicating confidence of interactions.

**Figure 8 ijms-26-07746-f008:**
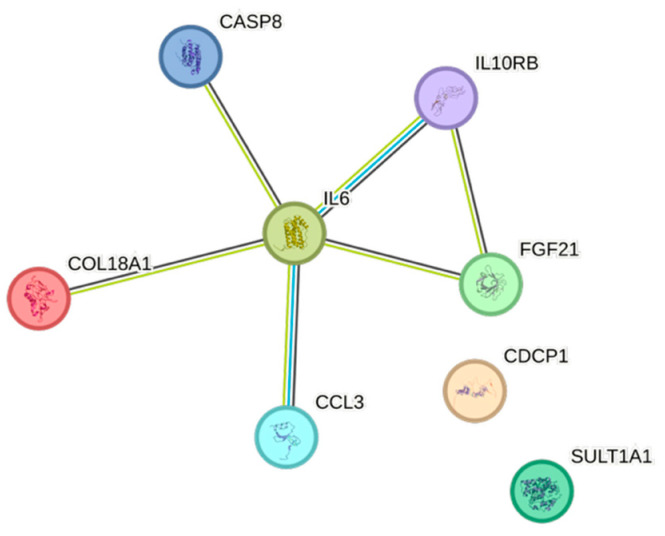
This figure represents a network of interactions among CCGFs, demonstrating significant associations between circulating cytokine levels and remnants. The nodes (CCGFs) are connected by edges of varying thickness, indicating confidence of interactions.

**Figure 9 ijms-26-07746-f009:**
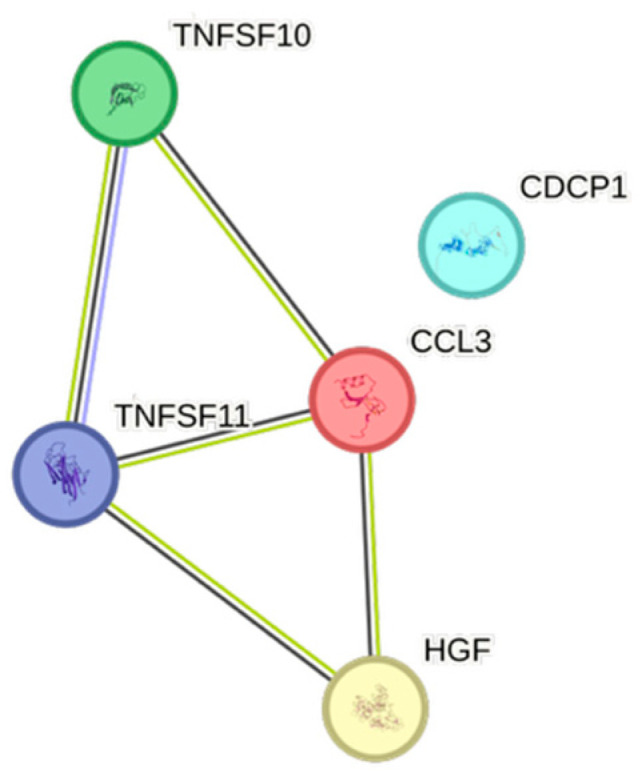
This figure represents a network of interactions among CCGFs, demonstrating significant associations between circulating cytokine levels and non-HDL. The nodes (CCGFs) are connected by edges of varying thickness, indicating confidence of interactions.

**Figure 10 ijms-26-07746-f010:**
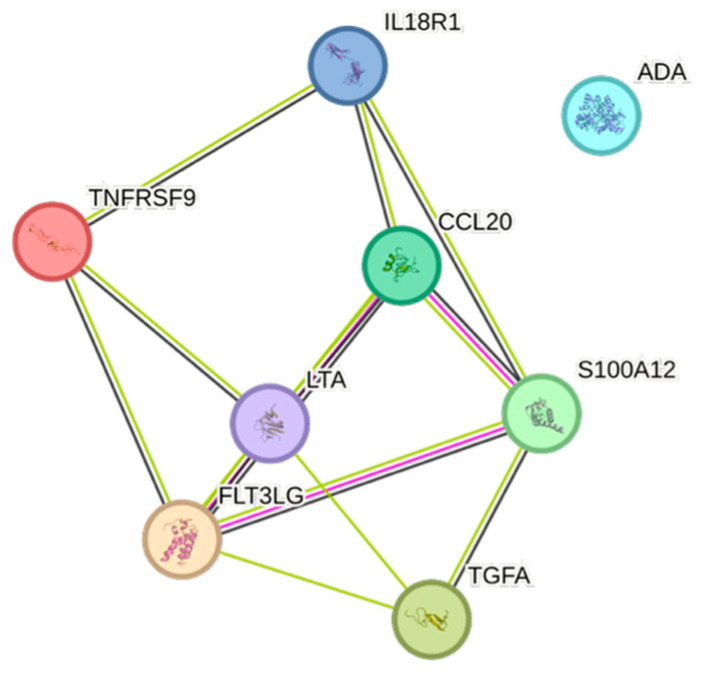
This figure represents a network of interactions among CCGFs, demonstrating significant associations between circulating cytokine levels and age. The nodes (CCGFs) are connected by edges of varying thickness, indicating confidence of interactions.

**Figure 11 ijms-26-07746-f011:**
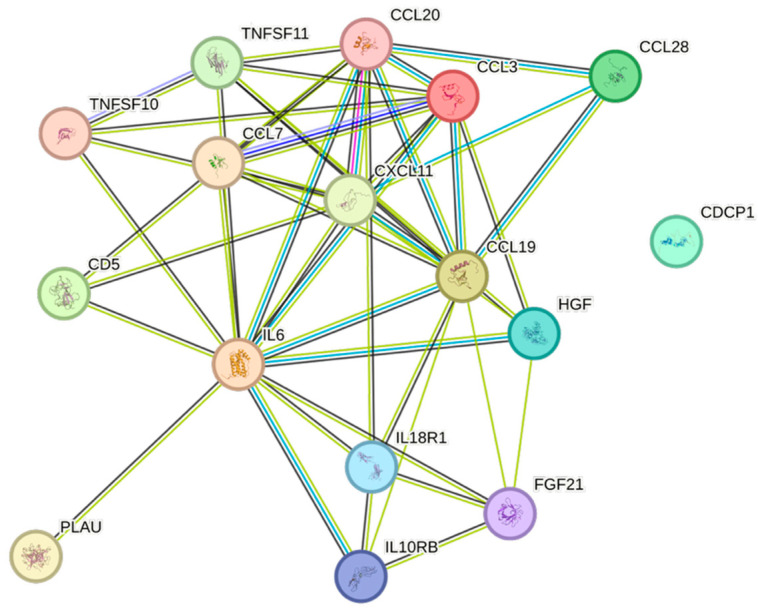
This figure represents a network of interactions among CCGFs, demonstrating significant associations between circulating cytokine levels and weight. The nodes (CCGFs) are connected by edges of varying thickness, indicating confidence of interactions.

**Figure 12 ijms-26-07746-f012:**
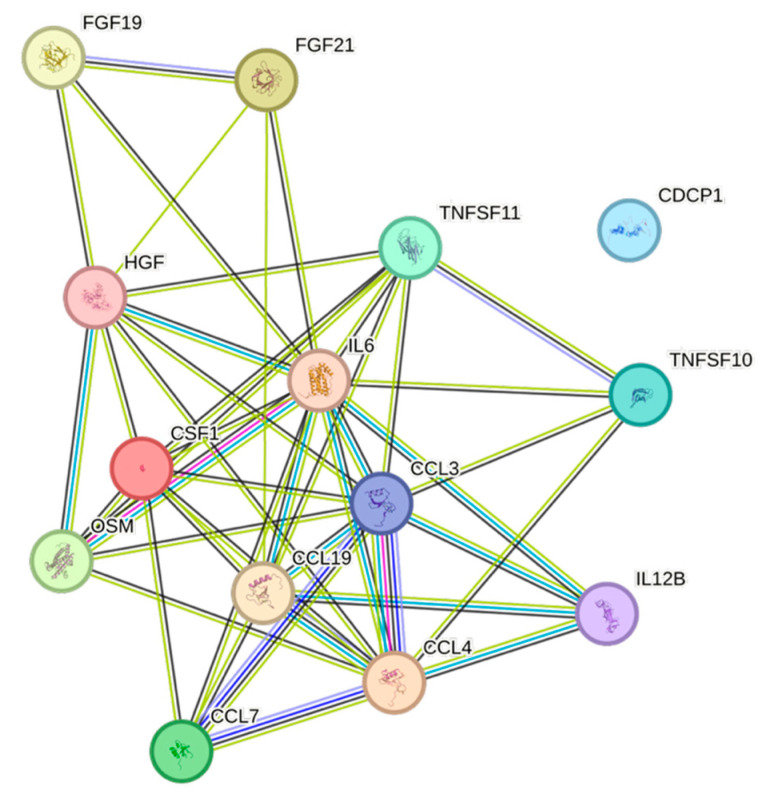
This figure represents a network of interactions among CCGFs, demonstrating significant associations between circulating cytokine levels and BMI. The nodes (cytokines) are connected by edges of varying thickness, indicating confidence of interactions.

**Figure 13 ijms-26-07746-f013:**
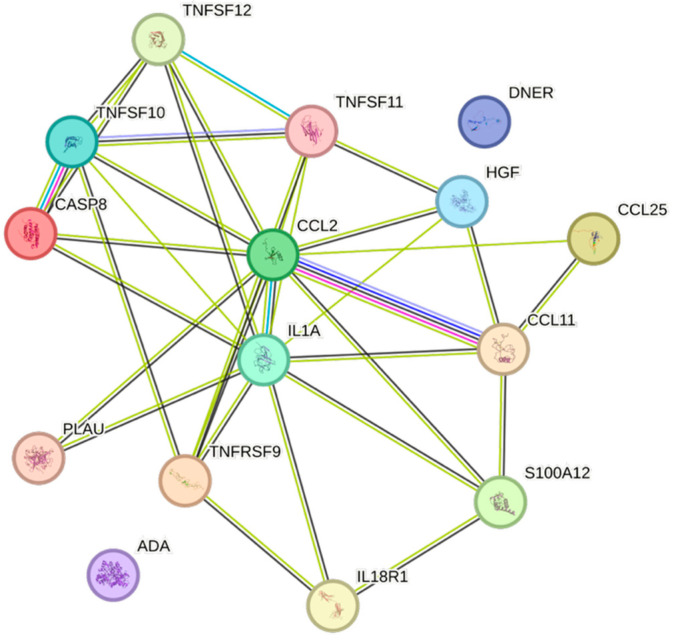
This figure represents a network of interactions among CCGFs, demonstrating significant associations between circulating cytokine levels and gender. The nodes (CCGFs) are connected by edges of varying thickness, indicating confidence of interactions.

**Table 1 ijms-26-07746-t001:** This table summarizes the prevalence of the CCGFs in our study cohort. Prevalences are given in absolute numbers. Abbreviations are explained in Appendix A, where UniprotIDs are also given.

Cytokine	Prevalence	Cytokine	Prevalence	Cytokine	Prevalence
TNFSF10	10	CASP-8	3	CGCP1	1
CCL3	8	COL18A1	3	CXCL11	1
TNFSF11	7	TNFSF12	3	CX3CL1	1
CDCP1	6	CSF-1	2	Flt3L	1
IL6	6	SULT1A1	2	IL-17C	1
FGF-21	5	CD5	2	OSM	1
S100A12	5	TGF-alpha	2	CCL4	1
TNFRSF9	5	DNER	2	IL-12B	1
PLAU	5	ADA	2	IL-15RA	1
HGF	5	CCL11	2	IL7	1
CCL7	5	CCL20	2	CD6	1
IL-18R1	4	CCL25	2	TNF	1
CCL19	4	MMP-10	1	Beta-NGF	1
IL-10RB	3	IL-1 alpha	1	LTA	1
FGF-19	3	CCL2	1	CCL28	1

**Table 2 ijms-26-07746-t002:** Number of CCGFs below the assay’s standard points.

	CCGF	UniProt N°	Olink Assay ID	N° Below Limit of Quantification
Olink Target 96 Inflammation	IL8	P10145	OID00471	0
Olink Target 96 Inflammation	VEGFA	P15692	OID00472	0
Olink Target 96 Inflammation	CD8A	P01732	OID05124	0
Olink Target 96 Inflammation	MCP-3	P80098	OID00474	74
Olink Target 96 Inflammation	GDNF	P39905	OID00475	1
Olink Target 96 Inflammation	CDCP1	Q9H5V8	OID00476	0
Olink Target 96 Inflammation	CD244	Q9BZW8	OID00477	0
Olink Target 96 Inflammation	IL7	P13232	OID00478	1
Olink Target 96 Inflammation	OPG	O00300	OID00479	0
Olink Target 96 Inflammation	LAP TGF-beta-1	P01137	OID00480	0
Olink Target 96 Inflammation	uPA	P00749	OID00481	0
Olink Target 96 Inflammation	IL6	P05231	OID00482	0
Olink Target 96 Inflammation	IL-17C	Q9P0M4	OID00483	12
Olink Target 96 Inflammation	MCP-1	P13500	OID00484	0
Olink Target 96 Inflammation	IL-17A	Q16552	OID00485	55
Olink Target 96 Inflammation	CXCL11	O14625	OID00486	0
Olink Target 96 Inflammation	AXIN1	O15169	OID00487	0
Olink Target 96 Inflammation	TRAIL	P50591	OID00488	0
Olink Target 96 Inflammation	IL-20RA	Q9UHF4	OID00489	125
Olink Target 96 Inflammation	CXCL9	Q07325	OID00490	0
Olink Target 96 Inflammation	CST5	P28325	OID00491	0
Olink Target 96 Inflammation	IL-2RB	P14784	OID00492	57
Olink Target 96 Inflammation	IL-1 alpha	P01583	OID00493	142
Olink Target 96 Inflammation	OSM	P13725	OID00494	0
Olink Target 96 Inflammation	IL2	P60568	OID00495	155
Olink Target 96 Inflammation	CXCL1	P09341	OID00496	0
Olink Target 96 Inflammation	TSLP	Q969D9	OID00497	128
Olink Target 96 Inflammation	CCL4	P13236	OID00498	0
Olink Target 96 Inflammation	CD6	P30203	OID00499	0
Olink Target 96 Inflammation	SCF	P21583	OID00500	0
Olink Target 96 Inflammation	IL18	Q14116	OID00501	0
Olink Target 96 Inflammation	SLAMF1	Q13291	OID00502	0
Olink Target 96 Inflammation	TGF-alpha	P01135	OID00503	0
Olink Target 96 Inflammation	MCP-4	Q99616	OID00504	0
Olink Target 96 Inflammation	CCL11	P51671	OID00505	0
Olink Target 96 Inflammation	TNFSF14	O43557	OID00506	0
Olink Target 96 Inflammation	FGF-23	Q9GZV9	OID00507	0
Olink Target 96 Inflammation	IL-10RA	Q13651	OID00508	52
Olink Target 96 Inflammation	FGF-5	P12034	OID00509	70
Olink Target 96 Inflammation	MMP-1	P03956	OID00510	0
Olink Target 96 Inflammation	LIF-R	P42702	OID00511	0
Olink Target 96 Inflammation	FGF-21	Q9NSA1	OID00512	6
Olink Target 96 Inflammation	CCL19	Q99731	OID00513	0
Olink Target 96 Inflammation	IL-15RA	Q13261	OID00514	102
Olink Target 96 Inflammation	IL-10RB	Q08334	OID00515	0
Olink Target 96 Inflammation	IL-22 RA1	Q8N6P7	OID00516	142
Olink Target 96 Inflammation	IL-18R1	Q13478	OID00517	0
Olink Target 96 Inflammation	PD-L1	Q9NZQ7	OID00518	0
Olink Target 96 Inflammation	Beta-NGF	P01138	OID00519	155
Olink Target 96 Inflammation	CXCL5	P42830	OID00520	0
Olink Target 96 Inflammation	TRANCE	O14788	OID00521	0
Olink Target 96 Inflammation	HGF	P14210	OID00522	0
Olink Target 96 Inflammation	IL-12B	P29460	OID00523	0
Olink Target 96 Inflammation	IL-24	Q13007	OID00524	151
Olink Target 96 Inflammation	IL13	P35225	OID00525	134
Olink Target 96 Inflammation	ARTN	Q5T4W7	OID00526	130
Olink Target 96 Inflammation	MMP-10	P09238	OID00527	0
Olink Target 96 Inflammation	IL10	P22301	OID00528	0
Olink Target 96 Inflammation	TNF	P01375	OID05548	0
Olink Target 96 Inflammation	CCL23	P55773	OID00530	0
Olink Target 96 Inflammation	CD5	P06127	OID00531	0
Olink Target 96 Inflammation	CCL3	P10147	OID00532	0
Olink Target 96 Inflammation	Flt3L	P49771	OID00533	0
Olink Target 96 Inflammation	CXCL6	P80162	OID00534	0
Olink Target 96 Inflammation	CXCL10	P02778	OID00535	0
Olink Target 96 Inflammation	4E-BP1	Q13541	OID00536	0
Olink Target 96 Inflammation	IL-20	Q9NYY1	OID00537	157
Olink Target 96 Inflammation	SIRT2	Q8IXJ6	OID00538	0
Olink Target 96 Inflammation	CCL28	Q9NRJ3	OID00539	0
Olink Target 96 Inflammation	DNER	Q8NFT8	OID01213	0
Olink Target 96 Inflammation	EN-RAGE	P80511	OID00541	0
Olink Target 96 Inflammation	CD40	P25942	OID00542	162
Olink Target 96 Inflammation	IL33	O95760	OID00543	0
Olink Target 96 Inflammation	IFN-gamma	P01579	OID05547	0
Olink Target 96 Inflammation	FGF-19	O95750	OID00545	0
Olink Target 96 Inflammation	IL4	P05112	OID00546	116
Olink Target 96 Inflammation	LIF	P15018	OID00547	156
Olink Target 96 Inflammation	NRTN	Q99748	OID00548	149
Olink Target 96 Inflammation	MCP-2	P80075	OID00549	0
Olink Target 96 Inflammation	CASP-8	Q14790	OID00550	0
Olink Target 96 Inflammation	CCL25	O15444	OID00551	0
Olink Target 96 Inflammation	CX3CL1	P78423	OID00552	0
Olink Target 96 Inflammation	TNFRSF9	Q07011	OID00553	0
Olink Target 96 Inflammation	NT-3	P20783	OID00554	3
Olink Target 96 Inflammation	TWEAK	O43508	OID00555	0
Olink Target 96 Inflammation	CCL20	P78556	OID00556	0
Olink Target 96 Inflammation	ST1A1	P50225	OID00557	2
Olink Target 96 Inflammation	STAMBP	O95630	OID00558	0
Olink Target 96 Inflammation	IL5	P05113	OID00559	127
Olink Target 96 Inflammation	ADA	P00813	OID00560	0
Olink Target 96 Inflammation	TNFB	P01374	OID00561	0
Olink Target 96 Inflammation	CSF-1	P09603	OID00562	0

**Table 3 ijms-26-07746-t003:** Interquartile range (=IQR). Erythrocyte volume fraction (EVF), white blood cell count (WBC), platelet count (Plt), albumin (Alb), and creatinine (Crea).

		Valid N	Median	IQR
Sex		68% females		
Age	year	164	29	12
Weight	kg	164	72.9	17
BMI	kg/m^2^	164	24.3	5
Hb	g/L	164	134	16
EVF	%	164	41	4
WBC	×10^9^/L	164	5.5	2
Plt	×10^9^/L	164	239	66
Alb	g/L	164	42	4
Crea	micromol/L	164	67	17
Cortisol	nanomol/L	164	360	199

## Data Availability

The dataset used and analyzed during the current study is available from the corresponding author on reasonable request.

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
