# Peer review of "Significant Interplay Between Lipids, Cytokines, Chemokines, Growth Factors, and Blood Cells in an Outpatient Cohort"

_ijms, 2025, doi:10.3390/ijms26167746_

Round 1

Reviewer 1 Report

Comments and Suggestions for Authors

The authors have tested the correlations between plasma cytokines and lipid-related biomarkers and their relationships with circulating blood cell counts, in a cohort of 164 essentially healthy adults aged 18–44 years.  This study is a very minor extension of their previous article published in this journal few months ago.

In their previous article (Ref 21- Eriksson, L.B. et al. Int J Mol Sci 2025, 26, doi:10.3390/ijms26094065) the authors have used 165 samples, but it is not clear why one of the samples was removed from the current study.

Most of the biomarkers tested in the study are already reported in their previous publication (Ref 21) and only a few extra biomarkers are included in this study. The statistical analysis and the presentations are exactly in the same format as in Ref 21, in fact the statistical analysis description is exactly same as in Ref 21.

The description in the statistics section 4.6 “Coefficients of variation were analyzed using Spearman rank correlations” is not clear. Similarly, it is mentioned that “p-values were adjusted for multiplicity using the false discovery rate (FDR) approach [61]. Adjusted p-values less than 0.10, corresponding to an expected false discovery rate of at most 10%, were considered significant. Results were adjusted for multiple testing, with a significance threshold of p < 0.05”. How do we understand the cut-off used for thresholding? From the supplementary tables it appears that 10% significance use used as cut-off.

The “Cytokine values that were above or below the highest and lowest standard points were assigned the values of these standards”. The authors should provide the list with how many samples were replaced with these standards for each of the cytokines.

What are prevalences reported in the Table 1, are they absolute numbers or percentages? How are the calculated?

A table with the level of these cytokines between male and female participants with other sample characteristics in would have been useful.

The authors have provided a series of STRING interaction images for each of the association result. It is unclear how this is useful for the current study as we are not sure if the edge weights are derived from the current data or from the STRING database.

The partial correlations between the proteins in a 92x92 correlation matrix would be useful to assess the relationship among these proteins on top of the STRING images.

Importantly, the authors should report partial correlations after accounting for age and sex. A full table with partial correlation, actual p-values (not rounded to 3 decimal places) and the adjusted p-values in supplementary tables can be presented.

Author Response

Reviewer 1.

First of all, we would like to thank the reviewer for constructive criticism and pointing out some weaknesses in our manuscript, which found useful when improving our article.

The authors have tested the correlations between plasma cytokines and lipid-related biomarkers and their relationships with circulating blood cell counts, in a cohort of 164 essentially healthy adults aged 18–44 years.  This study is a very minor extension of their previous article published in this journal few months ago.

It is true that this study is related to our previous one (“Significant Associations Between Blood Cell Counts and Plasma Cytokines, Chemokines, and Growth Factors”), but here we show associations between biomarkers, especially between lipids and blood cells, as well as biomarkers versus cytokines. Therefore, we claim that this study adds new knowledge on the complex interactions between cytokines, lipids and some clinical characteristics, which are related to the potential inflammasome activation, even in essentially healthy young individuals.

In their previous article (Ref 21- Eriksson, L.B. et al. Int J Mol Sci 2025, 26, doi:10.3390/ijms26094065) the authors have used 165 samples, but it is not clear why one of the samples was removed from the current study.

The sample material for one of the samples was insufficient to allow the testing needed for the expanded tests performed for the present study. We only have cytokine values and blood cell data for this sample and as the reviewer stated these associations have already been published. This is the reason for excluding one sample.

Most of the biomarkers tested in the study are already reported in their previous publication (Ref 21) and only a few extra biomarkers are included in this study. The statistical analysis and the presentations are exactly in the same format as in Ref 21, in fact the statistical analysis description is exactly same as in Ref 21.

The Reviewer is quite right about this and we apologize for copy-paste. This para has been rephrased and the sentence on p < 0.05 has been omitted since it might have caused confusion. Otherwise, the meaning is the same as previously stated. The p<0.05 criteria is only used in the supplementary tables and we therefore have moved this to the supplementary table legends to reduce confusion.

The description in the statistics section 4.6 “Coefficients of variation were analyzed using Spearman rank correlations” is not clear. Similarly, it is mentioned that “p-values were adjusted for multiplicity using the false discovery rate (FDR) approach [61]. Adjusted p-values less than 0.10, corresponding to an expected false discovery rate of at most 10%, were considered significant. Results were adjusted for multiple testing, with a significance threshold of p < 0.05”. How do we understand the cut-off used for thresholding? From the supplementary tables it appears that 10% significance use used as cut-off.

This section is rephrased and now reads:

Coefficients of variation were analyzed using Spearman rank correlations in Statistica (StatSoft, Version 14; Tulsa, OK, USA). Absolute cytokine values exceeding the assay’s standard curve limits were set to the highest or lowest standard value, respectively. To account for the increased risk of false positives due to multiple comparisons, p-values were adjusted using the false discovery rate (FDR) approach [REF]. Adjusted p-values below 0.10, corresponding to an expected FDR of ≤10%, were considered statistically significant.

The “Cytokine values that were above or below the highest and lowest standard points were assigned the values of these standards”. The authors should provide the list with how many samples were replaced with these standards for each of the cytokines.

Olink/Scilife recommended initially that cytokine values that were above or below the highest and lowest standard points were assigned the values of these standards. We thus originally followed this recommendation and replaced the results with the value of the lowest standard point. They have later revised this recommendation and reports also values below the lowest standard point. As the company changed their recommendations, we revised our handling of results below the lowest standard point to also include these values in the statistical analysis. Unfortunately, we missed to revise the manuscript accordingly. We included the number of results below the lowest standard point for each of the Olink markers in an excel file.

This is now explained in the Results section (2.2) of our manuscript.

What are prevalences reported in the Table 1, are they absolute numbers or percentages? How are the calculated?

The prevalences denote the number of times that each CCGF was noted in the study cohort. It is now explained in the legend to Table 1 that this refers to absolute numbers.

A table with the level of these cytokines between male and female participants with other sample characteristics in would have been useful.

Please, see below. Correlation matrices are now given as supplementary tables 4 and 5.

The authors have provided a series of STRING interaction images for each of the association result. It is unclear how this is useful for the current study as we are not sure if the edge weights are derived from the current data or from the STRING database.

Our use of STRING images was explained in sections 4.5 and 5. Based on our findings, the STRING database enlightened our results in conjunction with present knowledge on this paradigm. Thus, the edge weights are calculated by the STRING database based on the associations found in our study.

The partial correlations between the proteins in a 92x92 correlation matrix would be useful to assess the relationship among these proteins on top of the STRING images.

Three correlation matrices (whole cohort, men, and women, respectively) are given as supplementary XL-files. We hope that the Reviewer will find these matrices sufficiently informative on the protein levels, not only in all participants, but also in male and female participants.

Importantly, the authors should report partial correlations after accounting for age and sex. A full table with partial correlation, actual p-values (not rounded to 3 decimal places) and the adjusted p-values in supplementary tables can be presented.

Please see our response above regarding the two sex-specific correlation matrices.

Reviewer 2 Report

Comments and Suggestions for Authors

In this interesting study, the authors analyzed samples from "healthy" subjects, considering lipid and cytokine biomarkers. This is a very interesting study, but some improvements are needed: 1- Include a table in the text that better illustrates the sample: age, BMI, weight, any physical activity, sex, medications, any supplements, blood chemistry parameters (triglycerides, total cholesterol, HDL, LDL, blood glucose, HbA, cortisol, etc.) and blood parameters (platelet count, white blood cell count, neutrophil count, etc.) 2- Given the importance of the inflammatory profile, as reported by the authors themselves, what is the relationship between NLRP3 and related cytokines in their samples? Considering the role of the inflammasome not only in established CVD but also in metabolic syndrome in general (insert and comment PMID: 28403789, PMID: 3220263)

Author Response

Reviewer 2.

In this interesting study, the authors analyzed samples from "healthy" subjects, considering lipid and cytokine biomarkers. This is a very interesting study, but some improvements are needed: 1- Include a table in the text that better illustrates the sample: age, BMI, weight, any physical activity, sex, medications, any supplements, blood chemistry parameters (triglycerides, total cholesterol, HDL, LDL, blood glucose, HbA, cortisol, etc.) and blood parameters (platelet count, white blood cell count, neutrophil count, etc.) 2- Given the importance of the inflammatory profile, as reported by the authors themselves, what is the relationship between NLRP3 and related cytokines in their samples? Considering the role of the inflammasome not only in established CVD but also in metabolic syndrome in general (insert and comment PMID: 28403789, PMID: 3220263)

Dear Reviewer,

Many thanks for your kind words on the importance of this study.

Improvements according to the Reviewer´s comments.

  1.  

We agree with the reviewer that the cohort should be better defined. However, according to GDPR (EU 2016/679), we were not allowed to collect individual data, which were not relevant for participation in the surgical procedure (e.g. physical activity). However, our ethical permit approved specific analyses of blood samples that were conducted at group levels, when significant.

A Table 3 (shown below is now inserted in the manuscript).

Valid  N

Median

IQR

Sex

68% females

Age

year

164

29

12

Weight

kg

164

72.9

17

BMI

kg/m2

164

24.3

5

Hb

g/L

164

134

16

EVF

%

164

41

4

WBC

x109/L

164

5.5

2

Plt

x109/L

164

239

66

Alb

g/L

164

42

4

Crea

micromol/L

164

67

17

Cortisol

nanomol/L

164

360

199

Table 3. Interquartile range (=IQR). Erythrocyte volume fraction (EVF), White blood cell count (WBC), platelet count (Plt), Albumin (Alb), and Creatinine (Crea).

  1.  

As the Reviewer remarks the innate immune system is crucial in most inflammatory reactions and events.

Hence, the following para is now added to the Discussion section in our manuscript:
“NLRP3 is a key component of the innate immune system and forms part of the inflammasome that has been implicated in chronic low-grade inflammation associated with metabolic syndrome, including obesity, insulin resistance, and type 2 diabetes [48]. NLRP3 elicits maturation of the cytokines IL-1β and IL-18 [49]. We noted that IL18R1, a receptor for IL-18, is negatively associated with age and gender, but positively associated with Apo B and weight.”

We did not fully understand the relevance of the suggested reference PMID: 3220263, which describes ageing of skin as a function of social and professional conditions. However, if there are relevant comments in this article (which is not an open access), we would be most grateful if the reviewer guides us in this context.

Round 2

Reviewer 2 Report

Comments and Suggestions for Authors

I agree new version